# Gut microbiota features associated with *Clostridioides difficile* colonization in dairy calves

Laurel E. Redding[1]*, Alexander S. Berry[2,3], Nagaraju Indugu[1], Elizabeth Huang[1], Daniel P. Beiting[3], Dipti Pitta[1]

**1** Department of Clinical Studies-New Bolton Center, School of Veterinary Medicine, University of Pennsylvania, Kennett Square, PA, United States of America, **2** Division of Gastroenterology, Hepatology, and Nutrition, Children's Hospital of Philadelphia, Philadelphia, PA, United States of America, **3** Department of Pathobiology, School of Veterinary Medicine, University of Pennsylvania, Philadelphia, PA, United States of America

\* lredding@vet.upenn.edu

**Data Availability Statement:** Data are available on the Sequence Read Archive (SRA) at https://www.ncbi.nlm.nih.gov/sra/?term=PRJNA644943.

## Abstract

Diarrheal disease, a major cause of morbidity and mortality in dairy calves, is strongly associated with the health and composition of the gut microbiota. *Clostridioides difficile* is an opportunistic pathogen that proliferates and can produce enterotoxins when the host experiences gut dysbiosis. However, even asymptomatic colonization with *C. difficile* can be associated with differing degrees of microbiota disruption in a range of species, including people, swine, and dogs. Little is known about the interaction between *C. difficile* and the gut microbiota in dairy calves. In this study, we sought to define microbial features associated with *C. difficile* colonization in pre-weaned dairy calves less than 2 weeks of age. We characterized the fecal microbiota of 80 calves from 23 different farms using 16S rRNA sequencing and compared the microbiota of *C. difficile*-positive (n = 24) and *C. difficile*-negative calves (n = 56). Farm appeared to be the greatest source of variability in the gut microbiota. When controlling for calf age, diet, and farm location, there was no significant difference in Shannon alpha diversity ($P = 0.50$) or in weighted UniFrac beta diversity ($P = 0.19$) between *C. difficile*-positive and–negative calves. However, there was a significant difference in beta diversity as assessed using Bray-Curtiss diversity ($P = 0.0077$), and *C. difficile*-positive calves had significantly increased levels of *Ruminococcus (gnavus group)* (*Adj. P* = 0.052), *Lachnoclostridium* (*Adj. P* = 0.060), *Butyricicoccus* (*Adj. P* = 0.060), and *Clostridium sensu stricto 2* compared to *C. difficile*-negative calves. Additionally, *C. difficile*-positive calves had fewer microbial co-occurrences than *C. difficile*–negative calves, indicating reduced bacterial synergies. Thus, while *C. difficile* colonization alone is not associated with dysbiosis and is therefore unlikely to result in an increased likelihood of diarrhea in dairy calves, it may be associated with a more disrupted microbiota.

**Funding:** LER was supported by the USDA National Institute of Food and Agriculture, AHDR Sec1433 Capacity program (Formula Funds).

**Competing interests:** The authors have declared that no competing interests exist.

## Introduction

Infectious diarrheal disease is one of the main causes of mortality in dairy calves [1,2], and calves less than 30 days of age are at highest risk of developing diarrhea [3,4]. Studies have shown that gut microbial composition is associated with gut health and the likelihood of diarrhea: reductions in microbial diversity are associated with an increased incidence of diarrhea [5], and the colonization of the calf gut with beneficial bacteria along with the decreased colonization of potential pathogens decreases the likelihood of calf diarrhea [6].

*Clostridioides difficile* is a spore-forming anaerobic, Gram-positive bacillus that is a significant enteric pathogen in many species of animals. Colonization with *C. difficile* has been shown to be associated with reduced gut microbial diversity and increased colonization of pathogenic bacteria in people [7,8], and we recently demonstrated a similar association in puppies [9]. Dairy calves, like the neonates of other species, are colonized with *C. difficile* at high rates, with reported prevalences ranging from 28–56% [10,11]. While there is some evidence that infection with *C. difficile* can result in diarrhea in calves [12], the effect of the asymptomatic colonization of calves on the gut microbiota is unknown. Given the crucial role of the gut microbiota in providing colonization resistance against pathogens that cause diarrhea [13,14], a better understanding of the effect of pathogens such as *C. difficile* on the calf gut microbiota is needed. The goal of this cross-sectional study was thus to define the gut microbiota features associated with *C. difficile* colonization in dairy calves and to define the effects of calf age, diet, and farm on the risk of colonization.

## Methods

### Sample collection

Fecal samples were manually collected from up to five randomly selected healthy calves less than three weeks of age from each of 23 dairy farms in Pennsylvania, Maryland and Delaware. Healthy calves were defined as non-diarrheic animals that were bright and alert and not characterized as sick by the farmer. Briefly, gentle rectal palpation was performed to stimulate defecation in each calf, and approximately 10 g of fecal material was captured directly from the rectum into a testing tube. Fecal material was transported to the lab within 1 hour of the collection time and stored at -80°C. Samples were thawed once for batch processing. The number of calves sampled was chosen to detect a minimum prevalence of *C. difficile* of 10% based on prevalences reported in the literature [12,15]. This study was approved by the Institutional Animal Care and Use Committee of the University of Pennsylvania (Protocol 806386).

### Detection of *C. difficile*

Individual fecal samples were tested for *C. difficile* using the Xpert *C. difficile* assay (Xpert CD assay; Cepheid, Sunnyvale, CA, USA) according to the manufacturer's instructions. This assay detects the cytotoxin gene (*tcdB*) and binary toxin genes (*cdtA* and *cdtB*). Additionally, the assay has a callout for ribotype NAP1/B1/027. While this assay has not been validated for use in bovine feces, the sensitivity and specificity of this assay in human stools are 93.5% (95% CI: 90.3–95.9%) and 94.0% (95% CI: 92.9–95.0%), respectively, according to the manufacturer, and 84.2% and 87.7%, respectively, in canine feces [16].

To rule out the possibility of colonization with non-toxigenic *C. difficile*, pooled fecal samples from each farm were also submitted for anaerobic culture as previously described [9,17]. Briefly, 0.5 g of formed fecal sample was mixed with 0.5 ml of 100% ethanol for 60 minutes at room temperature before being inoculated on Cycloserine-cefoxitin fructose modified agar (CCFA) (Remel™) or *Clostridium difficile* Selective Agar (BBL™) and Columbia CNA agar

(Thermo Fisher Scientific Remel Products). Inoculated plates and broth were incubated in BD Gas-Pak™ EZ container systems with BD BBL™ CO2 generators and BD BBL™ Gas Pak™ anaerobic CO2 indicators (Franklin Lakes, NJ) at 36˚C ± 2˚C under anaerobic growth conditions for seven days and checked for growth every other day. Suspect colonies were identified and isolated. Isolates were confirmed to be *C. difficile* by Maldi-TOF identification and/or RapID ANA II System (Thermo Fisher Scientific Remel Products).

## 16S rRNA sequencing

DNA was extracted from fecal samples using Qiagen PowerSoil DNA extraction kit. 16S rRNA sequencing was performed as described previously [9,18]. Briefly, the V4 region of the 16S rRNA gene was amplified using PCR, which was performed using Accuprime Pfx Supermix and custom primers for 2 minutes at 95˚C, 30 cycles of: 20 seconds at 95˚C, 15 seconds at 55˚C, 5 minutes at 72˚C; and 10 minutes at 72˚C [18]. PicoGreen quantification was used to normalize post-PCR products and AMPureXP beads were used to clean the combined pools. Libraries were quantified and sized using a Qubit 2.0 and Tapestation 4200, respectively. 250bp paired-end sequencing was performed using an Illumina MiSeq. Sequence data are publically available (SRA accession number: PRJNA644943). Positive and negative controls were confirmed after sequencing.

## Sequence data processing using QIIME2

The QIIME2 pipeline [19] was used to process and analyze 16S sequencing data. Samples were demultiplexed using q2-demux and denoised using Dada2 [20]. Sequences were aligned using maaft [21] and phylogenetic trees were reconstructed using fasttree [22]. Shannon alpha diversity, weighted UniFrac and Bray-Curtis beta diversity metrics were estimated using q2-core-metrics-diversity after samples were rarefied to 1941 reads per sample, and p-values were adjusted for multiple hypothesis testing using Benjamini-Hochberg (B-H) false discovery rate (FDR) corrections [23]. Taxonomy was assigned to sequences using q2-feature-classifier classify-sklearn [24] against the Silva reference database [25]. Taxa were collapsed to the genus level, when possible. OTUs with less than 1% average relative abundance across all samples were removed.

## Correlation analysis and differential feature selection

The association between *C. difficile* colonization and calf age and feeding type was assessed by univariable logistic regression to rule out the possibility of confounding by these variables. The association between *C. difficile* culture status and microbial alpha diversity of the gut was then determined using a linear mixed effects model as implemented in the lme4 package [26] in R where age was controlled for as a fixed effect and with farm and diet as random effects. The correlation between *C. difficile* culture status on gut microbiota beta diversity was determined using PERMANOVA as implemented in the vegan package [27] in R controlling for age, farm, and diet. Principal coordinate analyses were performed using the phyloseq package in R [28]. Differentially-abundant taxa were determined using LDA Effect Size (LEfSe) [29] and Analysis of Composition of microbiomes (ANCOM), and p-values were adjusted for multiple hypothesis testing using B-H FDR corrections in R. The Dice index [30] was used to determine the co-occurrence of bacterial genera. Boxplots and LEfSe plots were visualized using ggplot2 [31] and ggthemes.

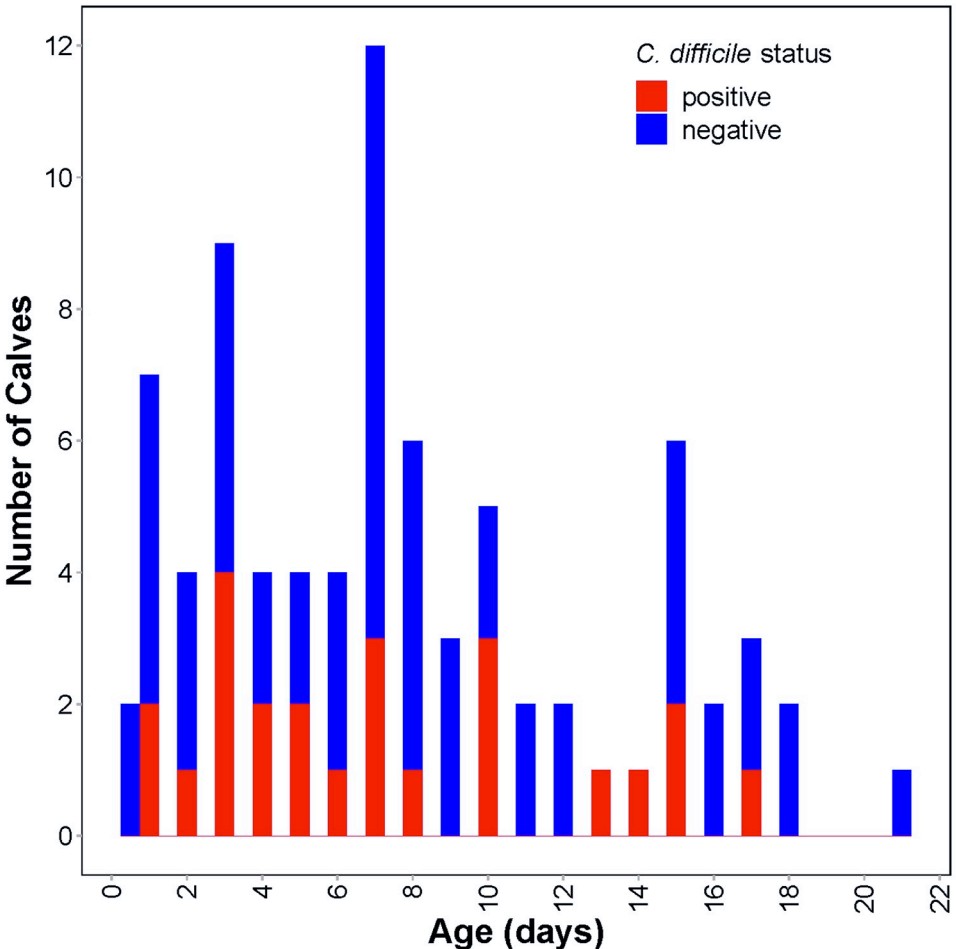

**Fig 1. Distribution of age and *C. difficile* colonization status in 92 pre-weaned Holstein dairy calves.**

## Results

### Subject characteristics and *C. difficile* status

Fecal samples were collected from a total of 92 Holstein calves from 23 farms. All calves appeared systemically healthy at the time of sampling and none had received antimicrobial therapy. The mean (SD) age of the calves was 7.0 (5.0) days. Thirty-six (35.6%) calves were fed waste milk, while the remaining calves were fed either colostrum or whole milk.

*C. difficile* was detected by qPCR in 28 calves (30.4%, 95% CI 21.2–40.9%) (**Fig 1**). Of the 28 samples that were positive for *C. difficile* on qPCR, 1 (3.6%) was positive for Toxin B only, 14 (50%) were positive for binary toxin only, and 13 (46.4%) were positive for both Toxin B and the binary toxin. None of the organisms were identified as the NAP1/B1/027 ribotype. On 14 farms, there were both *C. difficile*-positive and *C. difficile*-negative calves, whereas on the remaining farms, all of the calves were *C. difficile*-negative. There were no farms where all samples were qPCR-negative and the pooled sample from the farm was culture-positive. Neither calf age nor feeding of waste milk were significantly associated with the likelihood of detecting *C. difficile* among the calves (OR = 1.01, p = 0.805 and OR = 0.71, p = 0.493, respectively) (**Fig 1**).

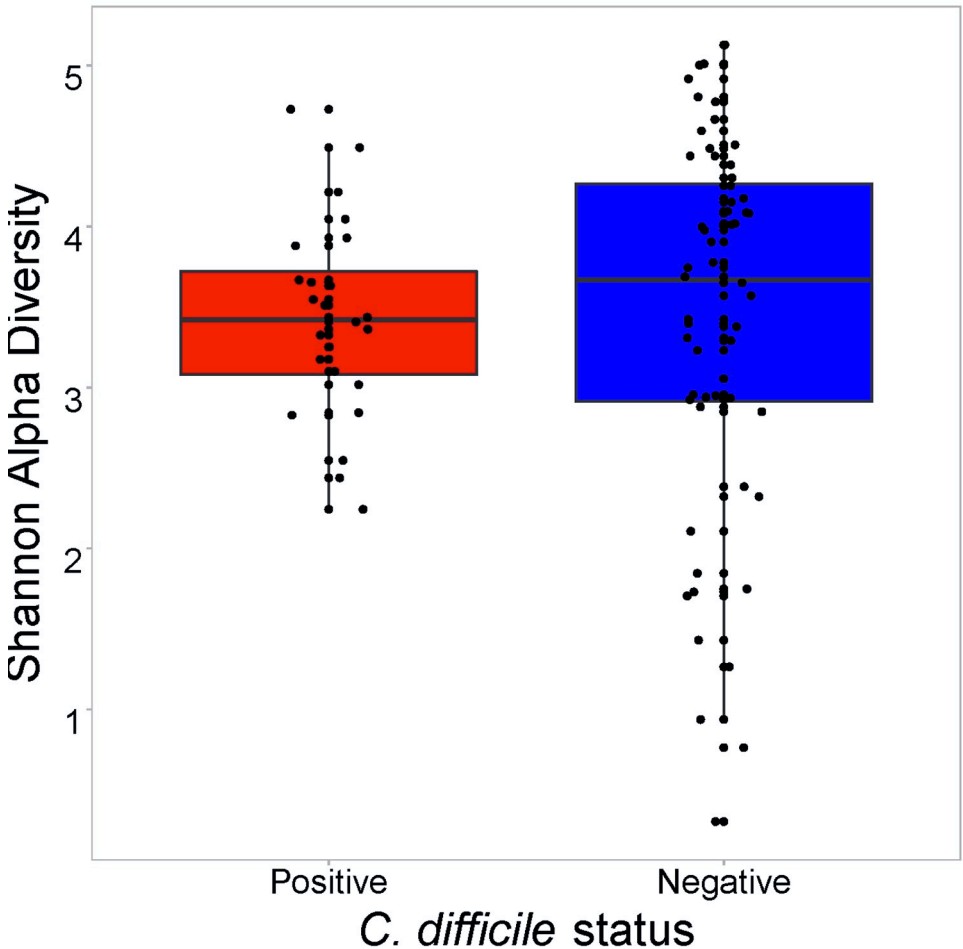

**Fig 2. Alpha diversity of the gut microbiome in 806 pre-weaned Holstein dairy calves by *C. difficile* colonization status.**

### Effect of *C. difficile* status on microbiota diversity

Sufficient fecal material was available for sequencing from 87 of the 92 calves. Microbiota community structure of the 87 calf fecal samples was assessed by sequencing and analyzing the V4 region of the 16S rRNA gene. Of the 87 that were sequenced, three were dropped from subsequent analyses because of low coverage and four were excluded because there was insufficient sample for qPCR analysis. Among the 80 remaining samples, 24 were positive for *C. difficile* by qPCR and 56 were negative.

The association between *C. difficile* status and Shannon alpha diversity was not significant ($P = 0.50$) when controlling for age, diet, and farm location (**Fig 2**). PERMANOVA was then used to test associations between *C. difficile* infection status and beta diversity of the gut microbiota. Farm location rather than *C. difficile* status explained most of the variation in gut microbiota composition across samples using both Bray-Curtis ($P = 1e\text{-}4$; $R^2 = 0.43$) and weighted UniFrac ($P = 1e\text{-}4$; $R^2 = 0.46$) beta diversity metrics (**Fig 3A and 3B**). Age and diet were not significantly associated with gut microbiota composition after controlling for farm ($P>0.1$). After controlling for farm, age, and diet, *C. difficile* status was significantly associated with Bray-Curtis beta diversity ($P = 0.0077$; $R^2 = 0.023$), explaining 2.3% of the variation in gut

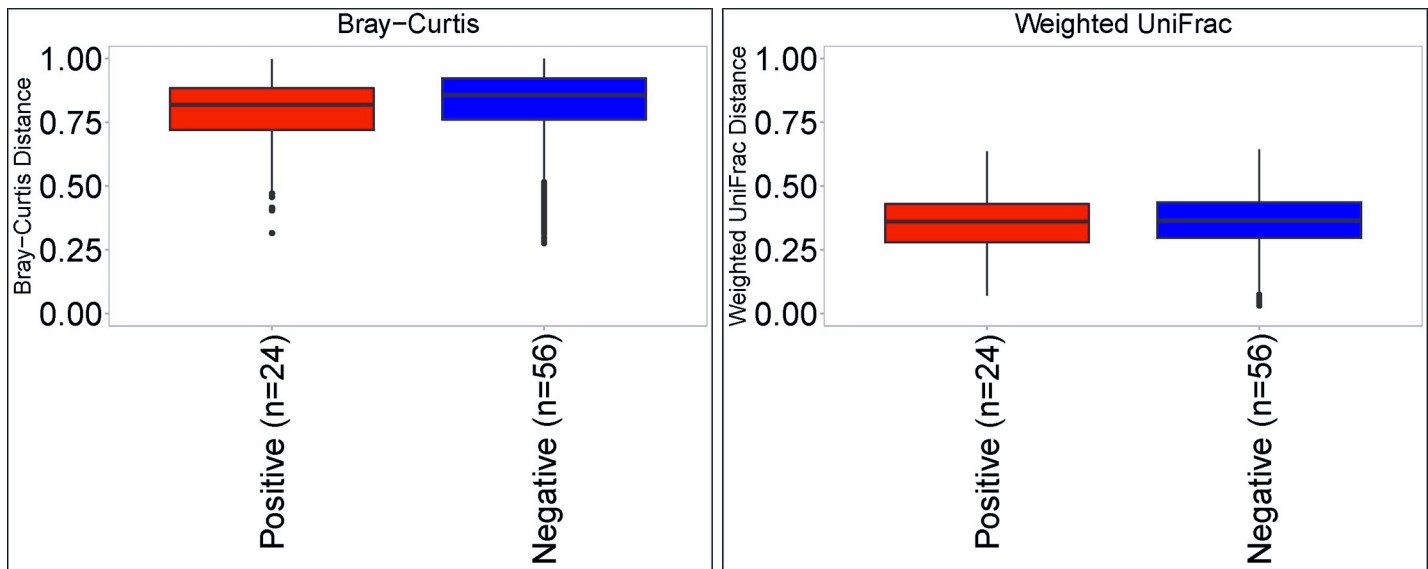

**Fig 3. Beta diversity of the gut microbiome in 80 pre-weaned Holstein dairy calves by *C. difficile* colonization status.** A. Bray-Curtis beta diversity. B. Weighted UniFrac.

microbiota composition. *C. difficile* status was not significantly associated with weighted UniFrac beta diversity ($P = 0.1934$; R2 = 0.013) after controlling for farm, age, and diet. Some clustering by farm and by *C. difficile* status within farms was apparent on principal coordinate analysis (**Figs 4 and S1**).

## Bacterial community composition

We next sought to determine the specific bacterial taxa associated with *C. difficile* colonization. At the phylum level, there were no significant differences between bacterial communities in *C. difficile*-positive and -negative samples (**Fig 5**). The Firmicutes phylum predominated (57.1% in *C. difficile*-positive samples and 51.4% in *C. difficile*-negative samples), followed by Proteobacteria (17.1% and 24.3%), Bacteroides (16.7% and 11.5%), and Actinobacteria (8.1% and 9.7%).

At the genus level, the only significant difference between *C. difficile*-positive and–negative samples by ANCOM occurred for Clostridioides. When considering LEFse analysis, there were four taxa among the 19 taxa with average relative abundance greater than 1% that were statistically significantly (*Adj. P*<0.1) associated with *C. difficile* status. *Ruminococcus (gnavus group)* (*Adj. P* = 0.052), *Lachnoclostridium* (*Adj. P* = 0.060), *Butyricicoccus* (*Adj. P* = 0.060), and *Clostridium* (*sensu stricto 2*) (*Adj. P* = 0.064) were all found in higher relative abundance among *C. difficile*-positive calves than in *C. difficile*-negative calves (**Fig 6**). While not statistically significantly different among the two groups, levels of *Lactobacillus*, *Megasphaera*, and *Streptococcus* were increased in *C. difficile*-positive samples, while levels of *Blautia*, *Fusobacterium*, *Tyzzerella*, *Enterobacteriaceae*, *Fecalibacterium*, *Dorea*, and *Collinsella* were decreased.

Co-occurrence analysis based on the Dice index showed that more co-occurrence of bacterial taxa appeared in the *C. difficile*-negative samples, with 1,488 (65.5%) highly (correlation coefficient>0.6) and significantly (p<0.01) correlated genera pairs. Most co-occurrences were among members of the Firmicutes phylum (1295, 55.0%). However, members of Firmicutes also showed high co-occurrence with Actinobacteria and Bacteroidetes. In the *C. difficile*-positive samples, there were fewer highly co-occurring genera, with 830 (73.3%) highly and

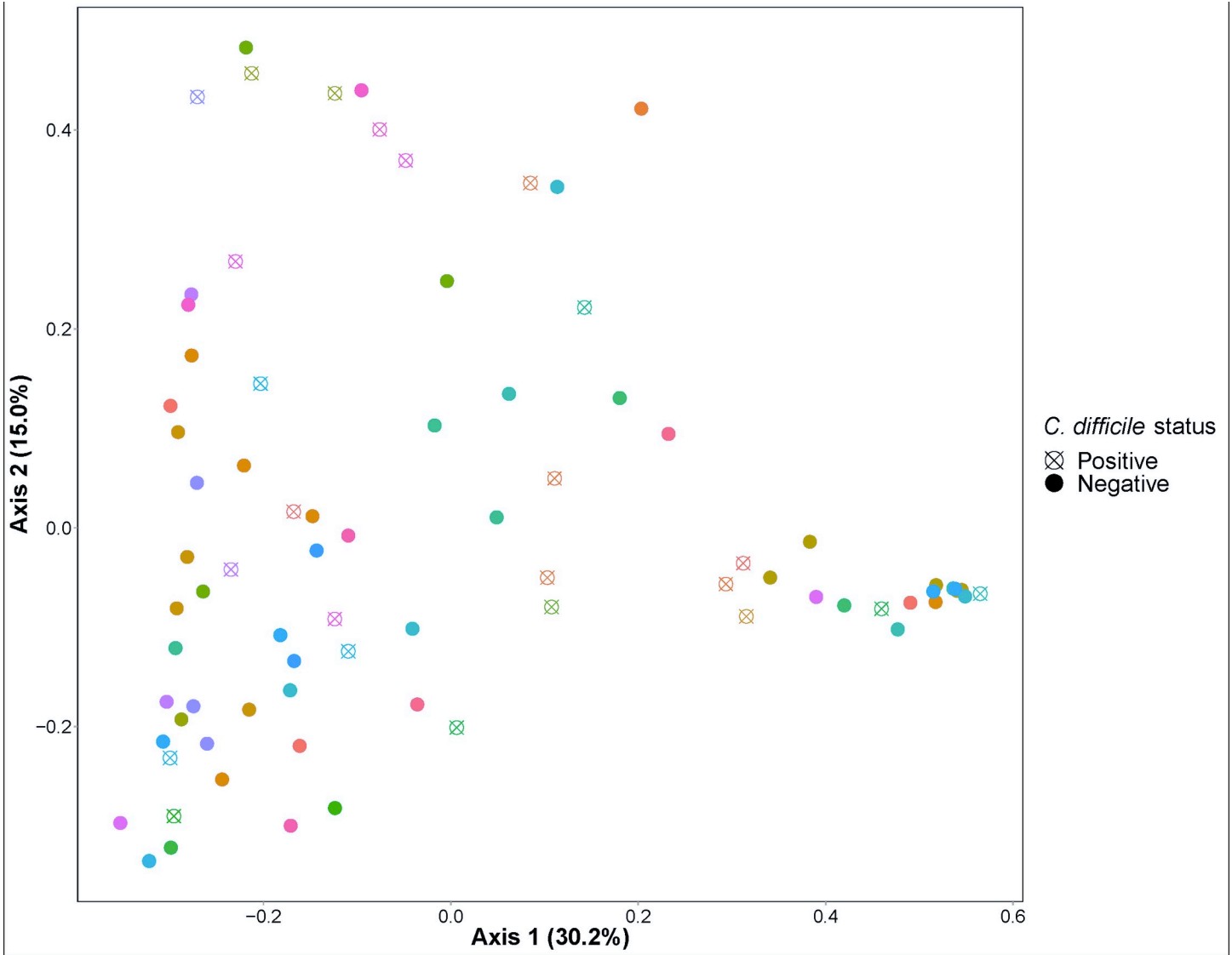

**Fig 4. Bray-Curtis principal coordinate analysis (PCoA) of fecal samples from 80 pre-weaned dairy calves by *C. difficile* colonization status and by farm.**

significantly correlated genera pairs. When only considering taxa with levels of relative abundance greater than 1%, there were no significant differences in co-occurrence patterns (**Fig 7**).

## Discussion

In this study, we characterized microbial features associated with asymptomatic *C. difficile* colonization in dairy calves. While the role of *C. difficile* in calf diarrhea remains equivocal [12], exploring the association between this pathogen and the gut microbiota is important for understanding factors that affect gut health and enteric diseases. A number of studies have examined the epidemiology of *C. difficile* in animals of veterinary importance, but the association between the microbiota and *C. difficile* is only beginning to be explored in dogs [9], horses [32], and pigs [33]. Notably, in pigs, the presence of *C. difficile* is associated with significantly reduced microbial diversity and increased levels of enteropathogens associated with neonatal diarrhea [33].

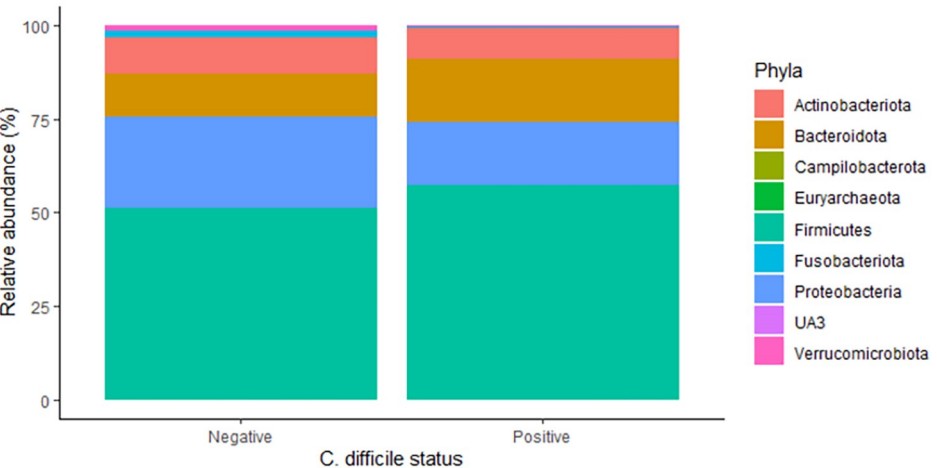

**Fig 5. Distribution of bacterial phyla by *C. difficile* status in fecal samples from 80 pre-weaned dairy calves.** The nine most abundant phyla are displayed.

Unsurprisingly, as in other studies [34–36], we found that the farm was the source of most of the variation in gut microbiota composition. However, even among calves from the same farm, there was variability in both *C. difficile* colonization status and gut microbial diversity, suggesting, as have other studies [35,37], that the farm environment is only one of many competing influencers of the developing calf gut microbiota. Neither diet nor age were significantly associated with microbiota composition when controlling for farm, but this is almost certainly due to the small sample size within each farm and the lack of within-farm variability in factors such as diet. When controlling for age, diet, and farm, we noted a significant difference in beta diversity between *C. difficile*-positive and *C. difficile*-negative fecal samples when considering the Bray-Curtis metric but not the unweighted UniFrac metric. While both of these metrics are weighted by abundance, the latter metric weighs diversity by phylogenetic relationship. Thus the lack of a significant difference when considering the weighted UniFrac metric suggests that, while there may be a significant difference in the composition of microbial communities, the differentially-abundant microbes might be closely related to one another. Indeed, all four genera identified as differentially-abundant by LEfSe are members of the *Clostridia* class, with two belonging to the *Clostrideaceae* family.

While the lack of a consistent difference in alpha and beta diversity between *C. difficile*-positive and *C. difficile*-negative samples suggests that the effect of *C. difficile* colonization on the gut microbiota of calves is minimal, other findings suggest that *C. difficile* colonization is associated with a more disrupted (i.e., altered)–but not dysbiotic (i.e., imbalanced and impaired)–gut microbiota. *C. difficile* colonization was preferentially associated with certain bacterial taxa of the class *Clostridia* that do have associations with dysbiosis. Notably, the overrepresentation of *Ruminococcus gnavus* and *Lachnoclostridia* in *C. difficile*-positive calves point to the possibility of an underlying imbalance in the gut microbiota. *R. gnavus*, a Gram-positive anaerobe that is typically found in the gut of over 90% of healthy people at relative abundances less than 0.1%, has been robustly associated with inflammatory dysbiotic conditions such as Crohn's disease [38–40], allergic airway disease [41], eczema [42], and spondyloarthritis [43]. Dramatic blooms of *R. gnavus* occur in patients experiencing flares of inflammatory bowel disease, with relative abundance levels that can peak at 69% of the gut microbiota [40]. Notably, this association appears to occur across species, as the gut microbiomes of both infants [7] and piglets [33] colonized with *C. difficile* also had increased relative levels of *Ruminococcus* species, including

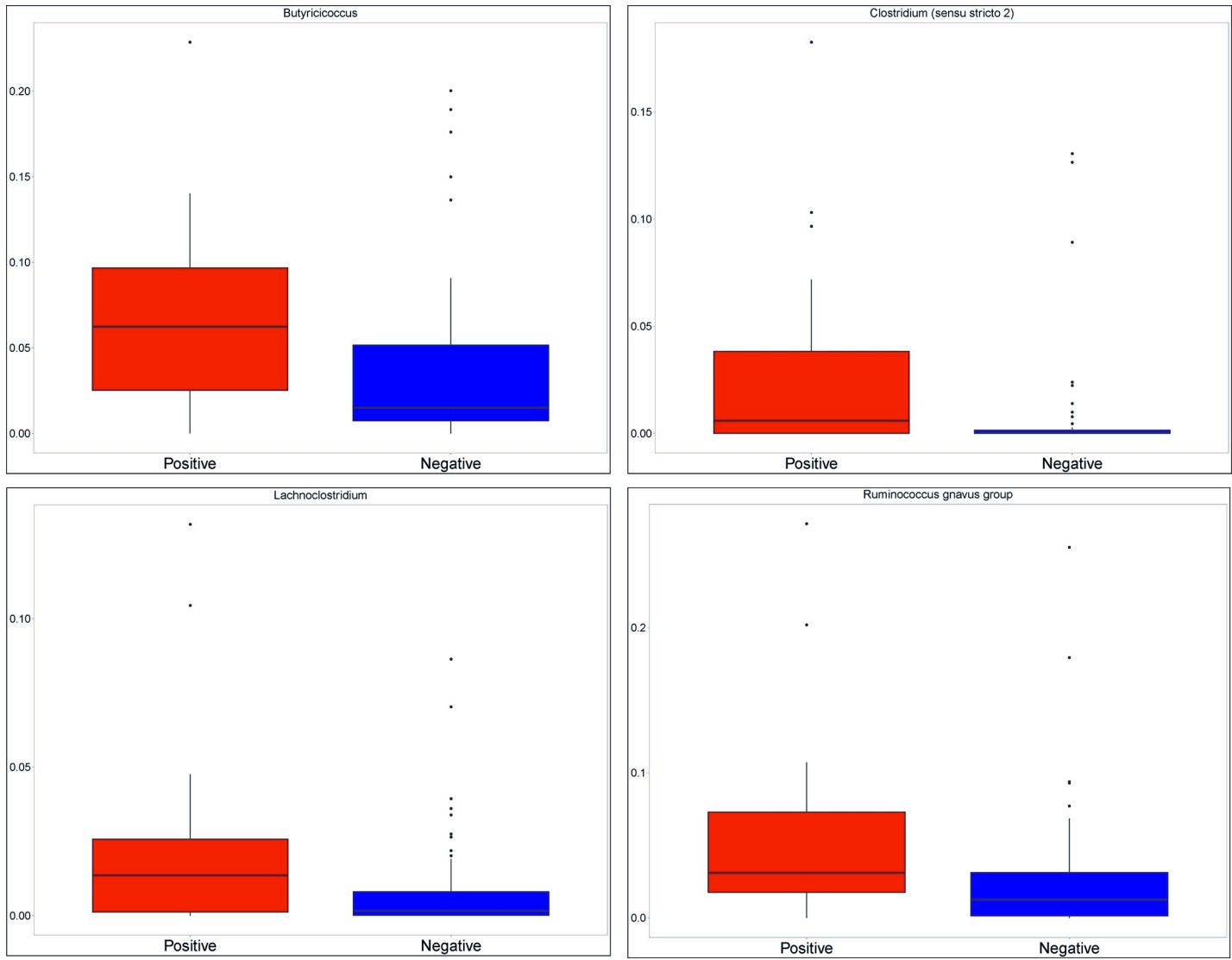

**Fig 6. Distribution of bacterial taxa that were found at higher levels in *C. difficile*-positive calves by *C. difficile* colonization status in 86 pre-weaned Holstein dairy calves.** A. *Butyricicoccus*. B. *Clostridium sensu stricto 2*. C. *Ruminococcus gnavus*. D. *Lachnoclostridium*.

*R. gnavus.* Additionally, *Ruminococcus* was one of six bacterial genera in the gut microbiota that predicted the occurrence of diarrhea in calves in another study [44]. The increased relative abundance of *Clostridium sensu stricto* and *Lachnoclostridia* in *C. difficile*-positive calves also points to the possibility of a less healthy gut environment. An increased relative abundance of *Clostridium sensu stricto*, which was also found in *C. difficile*-positive piglets [33], was associated with food allergies in infants [45] and diarrhea in piglets [46]. A tentative association between increased levels of *Lachnoclostridia* and neoplasia of the gastrointestinal tract has been identified in people [47,48]. While no such association has been explored in animals, the overrepresentation of this taxon in *C. difficile*-positive calves may be the result of a more disrupted gut microbiota. However, it is also important to note that the increased relative abundance of these taxa were only detected using LEfSe analysis and not ANCOM, which suggests that the association is likely relatively weak, as a stronger association would more likely have been consistently found across multiple metrics of differential relative abundance [49].

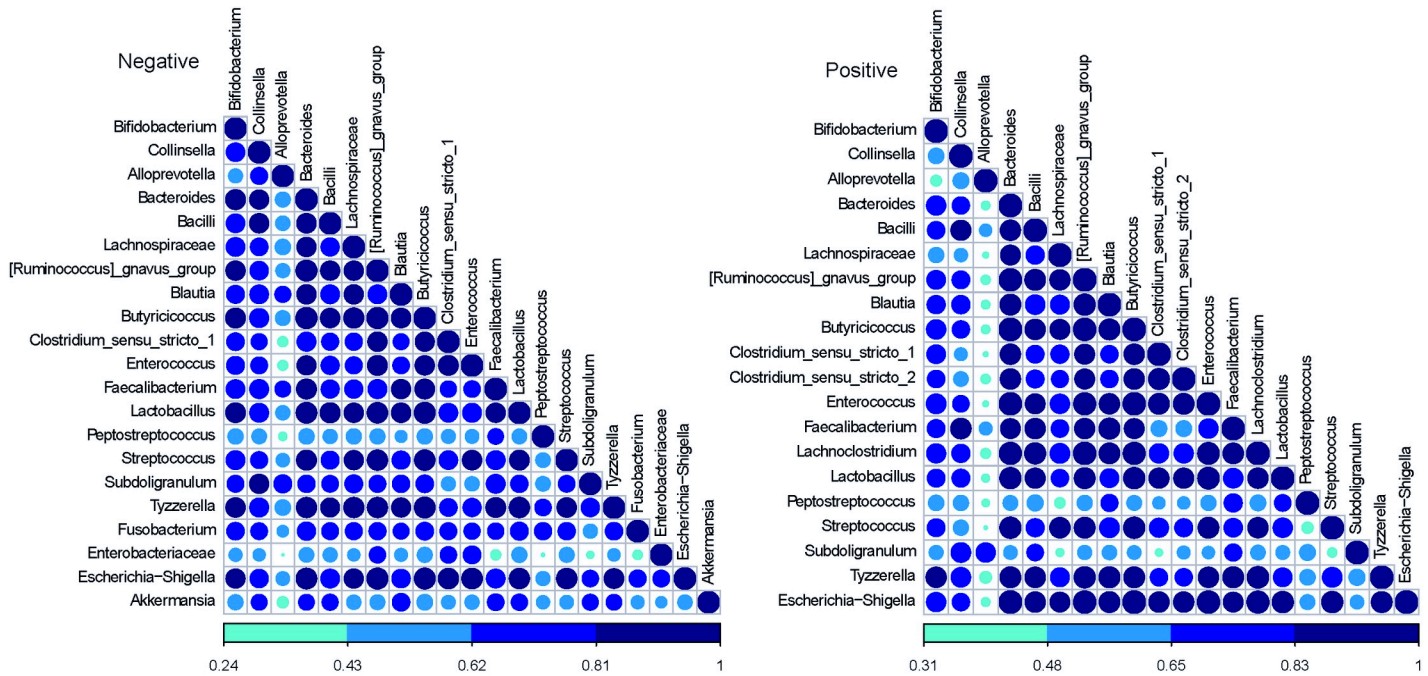

**Fig 7. Analysis of co-occurrence among microbial lineages scored using the Dice index by *C. difficile*-colonization status (positive and negative).** Dice indexes are shown as a heat map for all genera present at a level of relative abundance greater than 1% and with statistically significant (p<0.01) co-occurrence are shown as a heatmap. The degree of co-occurrence is shown by the color code at the bottom.

Certain bacterial taxa that predominate in healthy calves were found at lower (but not statistically significantly lower) levels in *C. difficile*-positive calves. Notably, *Fecalibacterium*, *Dorea*, *Enterobacteriaceae* and *Collinsella* are among the most abundant genera in healthy preweaned calves [50–53], and some of these taxa provide colonization resistance against *C. difficile* [8,54]. Their decreased relative abundance in *C. difficile*-positive calves is also reflective of a more disrupted gut microbiota. The decreased co-occurrence of bacterial taxa in *C. difficile*-positive calves compared to *C. difficile*-negative calves when considering all levels of relative abundance may also corroborate the notion of a slightly more disrupted gut microbiota in colonized calves. However, because the difference occurred only in rare taxa (relative abundance < 1%), this difference appears unlikely to result in dysbiosis.

One finding that is in contradiction to the general trend of *C. difficile* colonization being associated with disrupted microbiota is the increased relative abundance of *Butyricicoccus* in *C. difficile*-positive calves. In people, *Butyricicoccus* species of bacteria are generally found in *lower* levels in people colonized with *C. difficile* [55] or diagnosed with inflammatory bowel disease [56,57], and at higher levels in healthy dairy calves compared to calves with diarrhea [52,58]. It is unclear why the relative abundance of *Butyricioccus* occurred at higher levels in *C. difficile*-positive calves compared to *C. difficile*-negative calves. *Butyricicoccus* bacteria produce butyrate, an important nutrient source for gut colonocytes and a beneficial driver of the immunological maturation of the gut mucosa [59]. Moreover, this genus is one of the most abundant genera in dairy calves 7 days after birth [60]. The differential levels in calves compared to people with enteric disease may be due to species-specific patterns of development of the neonatal gut. Species-specific differences may also explain why *C. difficile* colonized calves had higher levels of *Clostridial* genera but colonized puppies had lower levels [9]. While rumen development is minimal in pre-weaned calves, they are nevertheless ruminants and thus have

fundamentally different enteric physiologies and microbial ecologies compared to true mono-gastric species.

Some limitations apply to this study. Heterogeneity in farm location, age, and diet across all of the sampled calves may have obscured features of the microbiota that would otherwise have been associated with *C. difficile* colonization. The cross-sectional nature of the study also precludes the possibility of drawing any conclusions about the duration of colonization and its effect on an already rapidly evolving gut microbiota. Moreover, while in some species the fecal microbiota is similar to that of the rectal mucosa [61–63], the fecal microbiota overall appears distinct from that of other parts of the GI tract, including the colon, where *C. difficile* resides [64–66]. The use of a qPCR assay to detect *C. difficile* that has not been validated in bovids could have resulted in false negatives or false positives. However, we believe this possibility to be minimal, because we have shown that the performance of this assay is relatively high in other species [16], and because pooled samples underwent anaerobic culture, which would have ruled out false negatives. Finally, because the qPCR assay only detects toxigenic C. difficile, we were unable to detect non-toxigenic *C. difficile*. It is likely that toxigenic and non-toxigenic *C. difficile* occupy a similar ecological niche and compete for similar resources within the gut microbiota; thus the presence of non-toxigenic *C. difficile* could account for the lack of a significant difference in alpha diversity and microbial composition between *C. difficile*-positive and *C. difficile*-negative calves. However, we believe this possibility to be unlikely, as there were no samples that were negative on qPCR but came from a farm where the pooled sample was positive for *C. difficile* on anaerobic culture.

## Conclusion

The greatest source of variability in the calf microbiota was the farm, and there were few or no statistically significant differences in alpha or beta diversity between *C. difficile*-positive and *C. difficile*-negative calves. *C. difficile* colonization thus does not appear to be associated with dysbiosis or with increased levels of enteropathogens that cause calf diarrhea. However, microbial community signatures–including increased relative abundance of bacterial taxa that that have been associated with dysbiotic states in other species and in people—suggest that the microbiota of *C. difficile*-colonized calves is more disrupted than that of non-colonized calves.

## Supporting information

**S1 Fig. Bray-Curtis principal coordinate analysis (PCoA) of fecal samples from 80 pre-weaned dairy calves.** Point shape identifies *C. difficile* colonization status and color denotes farm. 95% confidence intervals were drawn around samples collected in the same farm. (TIF)

## Author Contributions

**Conceptualization:** Laurel E. Redding, Alexander S. Berry, Nagaraju Indugu, Daniel P. Beiting, Dipti Pitta.

**Data curation:** Laurel E. Redding, Alexander S. Berry.

**Formal analysis:** Laurel E. Redding, Alexander S. Berry, Nagaraju Indugu, Dipti Pitta.

**Funding acquisition:** Laurel E. Redding.

**Investigation:** Laurel E. Redding, Alexander S. Berry, Elizabeth Huang.

**Methodology:** Laurel E. Redding, Alexander S. Berry, Nagaraju Indugu, Dipti Pitta.

**Project administration:** Laurel E. Redding, Daniel P. Beiting.

**Supervision:** Laurel E. Redding.

**Validation:** Laurel E. Redding, Alexander S. Berry.

**Visualization:** Laurel E. Redding, Alexander S. Berry, Nagaraju Indugu.

**Writing – original draft:** Laurel E. Redding, Alexander S. Berry.

**Writing – review & editing:** Laurel E. Redding, Alexander S. Berry, Dipti Pitta.

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
