## [Decision Letter · Decision Letter 0]

3 Sep 2021

PONE-D-21-14932

Gut microbiota features associated with Clostridioides difficile colonization in dairy calves

PLOS ONE

Dear Dr. Redding,

Thank you for submitting your manuscript to PLOS ONE. After careful consideration, we feel that it has merit but does not fully meet PLOS ONE’s publication criteria as it currently stands. Therefore, we invite you to submit a revised version of the manuscript that addresses the points raised during the review process.

PLEASE ADDRESS THE COMMENTS OF THE REVIEWERS IN THE REVISED VERSION.

We look forward to receiving your revised manuscript.

Kind regards,

Juan J Loor

Academic Editor

PLOS ONE

Journal Requirements:

3. Thank you for submitting the above manuscript to PLOS ONE. During our internal evaluation of the manuscript, we found significant text overlap between your submission and the following previously published works.

- https://doi.org/10.1016/j.anaerobe.2021.102353

- https://doi.org/10.1371/journal.pone.0215497

We would like to make you aware that copying extracts from previous publications, especially outside the methods section, word-for-word is unacceptable, even for works which you authored. In addition, the reproduction of text from published reports has implications for the copyright that may apply to the publications.

Please revise the manuscript to rephrase the duplicated text, cite your sources, and provide details as to how the current manuscript advances on previous work. Please note that further consideration is dependent on the submission of a manuscript that addresses these concerns about the overlap in text with published work.

Reviewers' comments:

Reviewer's Responses to Questions

**Comments to the Author**

1. Is the manuscript technically sound, and do the data support the conclusions?

Reviewer #1: Yes

Reviewer #2: Yes

2. Has the statistical analysis been performed appropriately and rigorously? 

Reviewer #1: Yes

Reviewer #2: Yes

3. Have the authors made all data underlying the findings in their manuscript fully available?

Reviewer #1: Yes

Reviewer #2: No

4. Is the manuscript presented in an intelligible fashion and written in standard English?

Reviewer #1: Yes

Reviewer #2: Yes

5. Review Comments to the Author

Reviewer #1: This is a very well written manuscript that presents interesting findings regarding C. diff in dairy calves and I only have minor comments to be addressed.

L62: Please provide additional details for the "manual" sample collection as this is not widely known by PlosOne readers (or provide a reference). Please include the amount of material that was collected, were samples processed the same day, shipping conditions, time in transport, etc. You state calves were less than 2-wk-of-age, but Figure 1 shows data from calves aged 21-22 days. Please clarify. Does the animal protocol number need to be included in the text?

Reviewer #2: Overall, the authors present a study with pertinence to the dairy industry and the results have potential implications for advancing our understanding of the fecal microbiota for benefiting calf health.

In reference to the technical soundness of this manuscript, the sequencing and analysis methods appear adequate; however, details are missing that are required for evaluation of this manuscript. Positive and negative controls should be listed for the DNA extraction kit and PCR; additionally, PCR specification details should be included. The authors should also substitute "microbiota" for "microbiome" throughout the manuscript, as they did not include any whole genome sequencing data. Also, the authors should comment on the accuracy of the Xpert kit used for C diff diagnosis with appropriate references included. In terms of the conclusions being supported by the data included, more clarity in the results section is needed. For example, one of the major findings is that farm location explained most of the variation of the fecal microbiota composition (ln 141-143). Figure 3 and 4 are both referenced in ln 141-143; however, Fig 3 doesn't illustrate this result of farm variability, as Bray-Curtis and unweighted UniFrac are shown for all C. diff + and C. diff - calves. The y axes for Figure 3a and 3b should also be labeled to improve clarity in this figure. Figure 4 should have a legend explaining the breakdown of colors and farm ID; clustering by farm should also be indicated on the PCA plot.

In terms of the methods/statistical analysis, sample size calculations should be included. Additionally, the method for selecting "healthy" calves should be included, specifically whether there were exclusion criteria. The authors should also be specific regarding whether they are referring to "relative abundance" when they indicate "abundance" or "levels" of taxa throughout the manuscript. Additionally, in line 127 the authors mention calculation of odds ratios for whether calf age or feeding waste milk was associated with the likelihood of detecting C. diff. This was not included in the methods section and this was not listed as a study objective. Also, the authors mention a mixed-effects ANOVA in line 136-141 that was not explained in the methods/statistical analysis section.

The authors did not include whether the raw sequence data had been submitted to NCBI SRA. If not available publicly, the raw abundance data should be listed as supplementary information for the manuscript.

Overall, the manuscript is written clearly; however, some errors need to be corrected. Please see individual line comments:

Ln 57: classify the study type (cross-sectional is stated in the discussion)

Ln 125: Consider revising to: "There were no farms where all samples were qPCR-negative and the pooled sample from the farm was positive". (“but” makes it sound like the pooled samples were positive)

Ln 127: Both factors (age and waste milk feeding) are not illustrated in Figure 1; consider revising

Ln 130: The authors should explain why 87 are listed instead of 92

Ln 133: Denominator is 80 for the calves that were sequenced and Figure 1 has 92 calves. Consider removing the “(Fig 1)” reference

Ln 136-141: move to methods section

Ln 141 and throughout: microbiota should be substituted for microbiota

Ln 142,143,146: revise R2 formatting to R^2

Ln 152-154: should be in a results/discussion section or moved to the discussion section

Ln 167-168: should be in the discussion or in a results/discussion section

Ln 169: Please clarify whether “all levels of abundance” refers to the resolution of taxa levels.

Ln 186-187 and throughout: Consider explaining why the fecal microbiota is representative of the gut microbiota or indicate why the complexity of the gut microbiota may not be captured with one fecal sample. Reference 31 evaluated the rumen microbiota, which may be distinct from findings in the fecal microbiota.

Ln 205: The distinction between disrupted and dysbiotic microbiota should be clarified.

Ln 223-255: Please add a citation/explanation as to why LEfSe analysis is weaker than ANCOM

Ln 239: The use of "they" pronoun is confusing. Consider revising to "it is unclear why the relative abundance of Butyricioccus was higher in..."

Ln 240-243: Please revise run-on sentence

Ln 245: genuses should be revised to genera

Ln 436: Microbiota should be substituted for microbiome. The results section indicates that 80 calves were sequenced. The correct denominator should be included with the breakdown of positive and negative C diff statuses in the figure legend. Whether 80 or 86 calves were sequenced should be consistent in all of the figure legends.

Ln 439: Same comments as above. The Y axis for Fig 3a and 3b is not labeled. Additionally, the number of positive C. diff and negative C. diff calves should be consistent in 3a and 3b. If not the authors should explain why they are different.

Ln 442: Explain in the figure legend how each color represents a different farm sampling location.

Figure 7: If there are only the taxa with mean RA > 1% included in the heat map, shouldn’t only 19 be listed based on Ln 160?. 21 are listed for C diff negative and 20 are listed for C diff positive calves. Please clarify.

6. PLOS authors have the option to publish the peer review history of their article (what does this mean?). If published, this will include your full peer review and any attached files.

Reviewer #1: No

Reviewer #2: No

---

## [Author Response · Author response to Decision Letter 0]

23 Sep 2021

Reviewer #1: This is a very well written manuscript that presents interesting findings regarding C. diff in dairy calves and I only have minor comments to be addressed.

AU: Thank you for your kind comments and your thorough review.

L62: Please provide additional details for the "manual" sample collection as this is not widely known by PlosOne readers (or provide a reference). Please include the amount of material that was collected, were samples processed the same day, shipping conditions, time in transport, etc. You state calves were less than 2-wk-of-age, but Figure 1 shows data from calves aged 21-22 days. Please clarify. Does the animal protocol number need to be included in the text?

AU: Manual sample collection was defined in the methods section, and additional information on transport and storage of processing. IACUC Protocol # was added. Thank you for pointing out the error with the age. The methods should say “3 weeks of age” or younger – we have corrected it. 

Reviewer #2: Overall, the authors present a study with pertinence to the dairy industry and the results have potential implications for advancing our understanding of the fecal microbiota for benefiting calf health.

In reference to the technical soundness of this manuscript, the sequencing and analysis methods appear adequate; however, details are missing that are required for evaluation of this manuscript. 

- Positive and negative controls should be listed for the DNA extraction kit and PCR; additionally, PCR specification details should be included. 

AU: We added details regarding positive and negative PCR controls in addition to PCR specifications to the methods section.

- The authors should also substitute "microbiota" for "microbiome" throughout the manuscript, as they did not include any whole genome sequencing data. 

AU: This was done. 

- Also, the authors should comment on the accuracy of the Xpert kit used for C diff diagnosis with appropriate references included. 

AU: Assay performance in humans and dogs has been added. Unfortunately, this assay has not been validated for use in bovine feces. This has also been added as a limitation in the discussion section.

- In terms of the conclusions being supported by the data included, more clarity in the results section is needed. For example, one of the major findings is that farm location explained most of the variation of the fecal microbiota composition (ln 141-143). Figure 3 and 4 are both referenced in ln 141-143; however, Fig 3 doesn't illustrate this result of farm variability, as Bray-Curtis and unweighted UniFrac are shown for all C. diff + and C. diff - calves. The y axes for Figure 3a and 3b should also be labeled to improve clarity in this figure. 

AU: We amended the phrase to say “Farm location rather than C. difficile status” so that the references to Fig. 3 and Fig 4 still apply (i.e., Fig 3 shows that C. difficile status was NOT significantly associated with beta diversity). Y-axis labels were added to both figures. 

- Figure 4 should have a legend explaining the breakdown of colors and farm ID; clustering by farm should also be indicated on the PCA plot.

AU: We added a legend to the main figure and added a supplemental figure that draws ellipses (95% confidence intervals) around samples from the same farm to show that samples tend to cluster by farm.. 

- In terms of the methods/statistical analysis, sample size calculations should be included. Additionally, the method for selecting "healthy" calves should be included, specifically whether there were exclusion criteria. 

AU: Information on both sample sizes and selection of healthy calves was added to the M&M section.

- The authors should also be specific regarding whether they are referring to "relative abundance" when they indicate "abundance" or "levels" of taxa throughout the manuscript. 

AU: We meant relative abundance throughout and have clarified in the manuscript. 

- Additionally, in line 127 the authors mention calculation of odds ratios for whether calf age or feeding waste milk was associated with the likelihood of detecting C. diff. This was not included in the methods section and this was not listed as a study objective. Also, the authors mention a mixed-effects ANOVA in line 136-141 that was not explained in the methods/statistical analysis section.

AU: This missing information was added to the methods section, and the reference to ANOVA was removed, as it was in fact the linear mixed effects model that was used, and not ANOVA.

- The authors did not include whether the raw sequence data had been submitted to NCBI SRA. If not available publicly, the raw abundance data should be listed as supplementary information for the manuscript.

AU: The raw sequence data have been deposited and an SRA number provided.

Overall, the manuscript is written clearly; however, some errors need to be corrected. Please see individual line comments:

Ln 57: classify the study type (cross-sectional is stated in the discussion)

AU: this was clarified.

Ln 125: Consider revising to: "There were no farms where all samples were qPCR-negative and the pooled sample from the farm was positive". (“but” makes it sound like the pooled samples were positive)

AU: Amended. Thank you for the suggestion.

Ln 127: Both factors (age and waste milk feeding) are not illustrated in Figure 1; consider revising

AU: Milk feeding was not illustrated in Figure 1 because it was a farm-level parameter, whereas age and C. difficile-status were individual calf parameters. 

Ln 130: The authors should explain why 86 are listed instead of 92. 

AU: the discrepancy in numbers has been explained in the results section and figure legends have been corrected. “87 samples were sequenced, but only 80 were included in the statistical analyses and figures due to sequencing depth and unknown C. diff status from PCR.”

Ln 133: Denominator is 80 for the calves that were sequenced and Figure 1 has 92 calves. Consider removing the “(Fig 1)” reference

AU: The Fig. 1 reference was removed from here. You are correct that Figure 1 reflects the entire cohort. 

Ln 136-141: move to methods section

AU : removed.

Ln 141 and throughout: microbiota should be substituted for microbiota

AU: Assuming you meant “microbiome”, we switched all instances of “microbiome” to “microbiota” 

Ln 142,143,146: revise R2 formatting to R^2

AU: Changed – thanks for pointing out this formatting error.

Ln 152-154: should be in a results/discussion section or moved to the discussion section

AU: This was rephrased.

Ln 167-168: should be in the discussion or in a results/discussion section

AU: This was rephrased.

Ln 169: Please clarify whether “all levels of abundance” refers to the resolution of taxa levels.

AU: this phrase was referring to the general levels only, but because it appeared confusing, we removed it.

Ln 186-187 and throughout: Consider explaining why the fecal microbiota is representative of the gut microbiota or indicate why the complexity of the gut microbiota may not be captured with one fecal sample. Reference 31 evaluated the rumen microbiota, which may be distinct from findings in the fecal microbiota.

AU: The lack of representativeness of the fecal microbiota for the rest of the GI tract has been noted as a limitation of this study in the discussion section.

Ln 205: The distinction between disrupted and dysbiotic microbiota should be clarified.

AU: short definitions of the two terms have been provided in this section.

Ln 223-255: Please add a citation/explanation as to why LEfSe analysis is weaker than ANCOM

AU: an explanation and a citation have been added: “which suggests that the association is likely relatively weak, as a stronger association would more likely have been consistently found across multiple metrics of differential abundance (46)”.

Ln 239: The use of "they" pronoun is confusing. Consider revising to "it is unclear why the relative abundance of Butyricioccus was higher in..."

AU: amended as suggested.

Ln 240-243: Please revise run-on sentence

AU: Revised as suggested.

Ln 245: genuses should be revised to genera

AU: revised.

Ln 436: Microbiota should be substituted for microbiome. The results section indicates that 80 calves were sequenced. The correct denominator should be included with the breakdown of positive and negative C diff statuses in the figure legend. Whether 80 or 86 calves were sequenced should be consistent in all of the figure legends.

AU: The substitution has been made, and corrections have been made to the figure legends. 

Ln 439: Same comments as above. The Y axis for Fig 3a and 3b is not labeled. Additionally, the number of positive C. diff and negative C. diff calves should be consistent in 3a and 3b. If not the authors should explain why they are different.

AU: Y axes were added, and figure legends were corrected to ensure consistency.

Ln 442: Explain in the figure legend how each color represents a different farm sampling location.

AU: a more complete legend was included, which should explain the color scheme clearly. 

Figure 7: If there are only the taxa with mean RA > 1% included in the heat map, shouldn’t only 19 be listed based on Ln 160?. 21 are listed for C diff negative and 20 are listed for C diff positive calves. Please clarify.

AU: Thank you for pointing this out. We believe this is a rounding error, and that the Dice index was for taxa with mean RA equal to or greater than 1%. The figure legend has been adapted accordingly.

---

## [Decision Letter · Decision Letter 1]

25 Nov 2021

Gut microbiota features associated with Clostridioides difficile colonization in dairy calves

PONE-D-21-14932R1

Dear Dr. Redding,

We’re pleased to inform you that your manuscript has been judged scientifically suitable for publication and will be formally accepted for publication once it meets all outstanding technical requirements.

Kind regards,

Juan J Loor

Academic Editor

PLOS ONE

Additional Editor Comments (optional):

Reviewers' comments:

Reviewer's Responses to Questions

**Comments to the Author**

1. If the authors have adequately addressed your comments raised in a previous round of review and you feel that this manuscript is now acceptable for publication, you may indicate that here to bypass the “Comments to the Author” section, enter your conflict of interest statement in the “Confidential to Editor” section, and submit your "Accept" recommendation.

Reviewer #1: All comments have been addressed

Reviewer #2: All comments have been addressed

2. Is the manuscript technically sound, and do the data support the conclusions?

Reviewer #1: Yes

Reviewer #2: Yes

3. Has the statistical analysis been performed appropriately and rigorously? 

Reviewer #1: Yes

Reviewer #2: Yes

4. Have the authors made all data underlying the findings in their manuscript fully available?

Reviewer #1: Yes

Reviewer #2: Yes

5. Is the manuscript presented in an intelligible fashion and written in standard English?

Reviewer #1: Yes

Reviewer #2: Yes

6. Review Comments to the Author

Reviewer #1: The authors have addressed my concerns.

There is still one minor inconsistency: Line 27 of the abstract should be corrected and changed to less than 3 weeks of age. Once this has been done the paper is acceptable for publication.

Reviewer #2: The authors adequately answered all of the reviewer comments and made the necessary revisions. I think the manuscript is well-written in its current form.

7. PLOS authors have the option to publish the peer review history of their article (what does this mean?). If published, this will include your full peer review and any attached files.

Reviewer #1: No

Reviewer #2: No

---

## [Editor Report · Acceptance letter]

6 Dec 2021

PONE-D-21-14932R1 

Gut microbiota features associated with *Clostridioides difficile* colonization in dairy calves 

Dear Dr. Redding:

I'm pleased to inform you that your manuscript has been deemed suitable for publication in PLOS ONE. Congratulations! Your manuscript is now with our production department. 

Kind regards, 

on behalf of

Dr. Juan J Loor 

Academic Editor

PLOS ONE